# Color Rendering Index over 95 Achieved by Using Light Recycling Process Based on Hybrid Remote-Type Red Quantum-Dot Components Applied to Conventional LED Lighting Devices

**DOI:** 10.3390/nano13182560

**Published:** 2023-09-15

**Authors:** Eunki Baek, Boseong Kim, Sohee Kim, Juyeon Song, Jaehyeong Yoo, Sung Min Park, Jong-Min Lee, Jae-Hyeon Ko

**Affiliations:** Nano Convergence Technology Center, School of Semiconductor∙Display Technology, Hallym University, Chuncheon 24252, Gangwon-do, Republic of Koreajmlee@hallym.ac.kr (J.-M.L.)

**Keywords:** quantum dot, LED, color rendering index, remote, quantum-dot component

## Abstract

Red color conversion materials have often been used in conventional white LEDs (light-emitting diodes) to enhance the insufficient deep-red component and thus improve the color-rendering property. Quantum dots (QDs) are one of the candidates for this due to their flexibility in controlling the emission wavelength, which is attributed to the quantum confinement effect. Two types of remote QD components, i.e., QD films and QD caps, were prepared and applied to conventional white LED illumination to improve the color-rendering properties. Thanks to the red component near 630 nm caused by the QD components, the color rendering indices (CRIs) of both Ra and R9 could be increased to over 95. It was found that both the diffusing nature of the reflector and the light recycling process in the vertical cavity between the bottom reflector and the top optical films play important roles in improving the color conversion efficiency of remote QD components. The present study showed that the proper application of remote QDs combined with a suitable optical cavity can control the correlated color temperature of the illumination over a wide range, thus realizing different color appearances of white LED illumination. In addition, a high CRI of over 95 could be achieved due to the sufficient excitation from fewer QDs, due to the strong optical cavity effect.

## 1. Introduction

Lighting technology is undergoing a revolutionary change based on white LEDs (light-emitting diodes). A new way of producing white light could be realized by combining blue LED chips with color conversion materials such as phosphors [1]. The most typical color conversion material for white LEDs is Ce-doped yttrium aluminum garnet (YAG, Y_3_Al_5_O_12_) [2], which is highly efficient and robust to environmental changes. However, it tends to lack a deep-red component, which is the main reason for its low color rendering index (CRI) [3]. The color-rendering properties of lighting are becoming increasingly important as people spend more time indoors following the outbreak of COVID-19. Several attempts have been made to improve the color-rendering property of white LEDs, such as using red and green phosphors simultaneously over the blue LEDs [4], applying red quantum dots (QDs) to conventional LED lighting devices [5,6], etc.

QDs are promising color conversion materials due to their high purity, color tunability and high quantum yield [7,8,9]. Due to their easy processibility and flexibility in the form factor, QDs have been adopted in backlighting for liquid crystal display (LCD) applications [10,11,12,13,14,15,16]. A high color gamut has been achieved for LCDs by using QD films in combination with blue LEDs, due to the high color purities of the primary lights emitted by red and green QDs. In contrast, the use of QDs for general illumination has been less studied. One reason may be the vulnerability of QDs to high temperatures and other ambient conditions such as humidity and oxygen. In this context, the remote-type design can be used to improve the long-term stability of QD-based lighting devices. Recent studies show that the CRI can be significantly increased by using remote QD components in various forms [17,18,19,20,21,22,23,24]. QDs immersed in polymers or glass composites have been studied by many groups for the past decades [25,26,27,28,29,30,31,32,33,34,35,36,37,38]. The most successful commercialization of remote QD components is QD films in backlighting for LCD applications. However, a careful design of the optical structure is required when incorporating remote QD components into conventional lighting to improve color conversion efficiency and to avoid color dispersion [39]. Otherwise, color conversion efficiency may become low, or light reabsorption may reduce the lighting efficiency.

In our previous studies, either QD films or QD caps have been incorporated into conventional white LED lighting devices to improve the color-rendering properties [39,40,41,42]. In particular, the position of each QD component has been optimized by evaluating the optical characteristics of the white LEDs. However, the interaction between the remote QD structure and other optical components has not been systematically investigated. The performance of optical components or color conversion materials is affected by the optical structure in addition to the optical properties of the materials that make up the lighting devices. For example, the gain factor of reflective polarizers used to increase the brightness of LCDs is critically dependent on the optical properties of the backlights, such as the stack of optical films, the reflective properties of the lower part of the backlight, etc. [43,44]. It shows that the process of light recycling in optical devices, such as backlights and illumination devices, plays an important role in improving optical performance.

The purpose of this study is to investigate the optimal configuration of remote QD components and their interactions with other optical properties, and thus to optimize lighting devices to achieve high color-rendering characteristics of conventional white LEDs. In particular, hybrid QD lighting devices consisting of both QD films and QD caps have been incorporated into the optical structure. The position, number, and shape of the two QD remote components, the sequence of optical films and the optical properties of the reflective surfaces have been optimized to achieve a high CRI of over 95.

## 2. Materials and Methods

A commercially available 6-inch, 15 W white LED lighting device (KE15DN61S57A1, Partner Co., Ltd., Gimpo, Korea) was adopted as a basic lighting fixture into which QD components were incorporated. A total of 72 white LEDs with an emitting area of 3.2 × 2.8 mm^2^ were arranged concentrically. Figure 1 shows the top view of one of the adopted designs, where the conventional white LED fixture can be seen together with the remote QD components, as described below. The upper and lower diameters of the lighting frame were 184 and 97 mm, respectively, resulting in an inclination angle of the side reflector of 131.5°. The CRI and the correlated color temperature (CCT) of the original white LED device without any QD components were 82.6 and 5626 K, respectively. The reflectance of the PCB (printed circuit board), on which the white LEDs were arranged, was 69%. A polycarbonate (PC) diffuser with a diameter of 147 mm and a thickness of 2 mm was used. The total transmittance and the haze of the diffuser were 69.9% and 99.2, respectively. The haze property was measured by using a haze meter (NDH-2000N, Nippon Denshoku, Tokyo, Japan).

CdSe/ZnS core–shell QDs with an average diameter of ~6 nm were fabricated by using the conventional hot injection method. The emitting wavelength of the red QDs was ~623 nm. The size of the QD nanoparticles was intentionally adjusted to ~6 nm to meet the requirement of the dominant wavelength of ~630 nm. This peak wavelength was chosen because it is preferable from both points of view of the efficacy and the CRI. If the wavelength is longer than this, the luminous efficiency of the photopic response becomes lower, resulting in low efficacy. If the wavelength is shorter than this, the color-rendering performance is expected to be poorer due to insufficient deep-red component.

Two remote red QD components were used: QD films and QD caps. The detailed fabrication process of the red QD films and QD caps has been reported before [39,41,42]. QD particles were mixed with irregular hollow silica (SG-HS40, Sukgyung AT Co., Ansan, Korea) of a size of ~40 ± 10 nm for dispersion. In order to fabricate the QD films, the QDs were mixed with triazine epoxy resin and coated on a PET (polyethylene terephthalate) substrate by using the roll-to-roll slot die method [39]. The QD concentration in the film was 7.5 wt%. At the initial stage, a QD concentration of 5.0 wt% was also prepared and tested in the lighting. The QD caps (QD concentration of 5 wt%) with the outer dimensions of 7.4 × 5.3 × 4.2 mm^3^ and two lateral thicknesses of 0.9 and 1.8 mm were fabricated by using a UV curing agent (Miracle UV Resin) and red QDs according to the method described in [41]. The upper surface has a rectangular opening with an area of 5.6 × 1.7 mm^2^. Figure 1 shows a photograph showing one design of the tested white LED devices, including the photoluminescence (PL) spectra of the QD films that were excited using an ultraviolet light of 365 nm. Some of the white LEDs were covered by QD caps, and the emitting area was surrounded by the QD stripe. The grey circular component on the right side is the diffuser. The dimensions and the photograph of the used QD cap are also shown in the same figure. Appendix A shows the optical components used in this study.

There were several factors in the optical configuration of the white LED lighting device: (1) the reflecting property of the reflector, whether the reflection was specular or diffuse, (2) whether the QD film was attached to the inclined side wall (which will be called “Ring-type”) or set vertically (which will be called “Wall-type”), and (3) the number of QD caps and their arrangement. The dimensions of the QD wall were 305 × 20 mm^2^. The QD ring was designed to cover the inclined side wall as schematically shown in Appendix A. Appendix A includes the photos of these two configurations. Figure 2 shows a cross-sectional view of the white LED lighting device. The top-view photographs of the diffuser film and the prism film are also shown in Figure 2. First, either a diffuse or a mirror reflector was laminated on the inclined side surface. The mirror reflector was an Al-coated one, while the diffuse reflector was a typical white PET film (See Appendix A). The ring-type and wall-type QD films can be seen in Appendix A, respectively. The 72 white LEDs were arranged in five concentric lines. Either the 12 white LEDs on the 3rd line or the 29 white LEDs on the 2nd and 4th lines were covered by the red QD caps, as shown in Appendix A.

The on-axis luminance was recorded using a 2D color analyzer (CA-2500A, Konica Minolta Co., Ltd., Tokyo, Japan). The emitting spectrum, the color coordinates, the CRI and the illuminance were measured by using an illuminance meter (CL-500A, Konica Minolta Co., Ltd., Tokyo, Japan). The distance between the center of the LED lighting and the color analyzer was 58 cm, while that between the lighting and the illuminance meter was 32 cm. The average CRI can be divided into two, Ra and Re; the former is the average of R1 to R8, while the latter is the average of R1 to R15, including R9, which is associated with the deep-red component. 

Table 1 shows several combinations of the optical films tried on the PC diffuser. Conventional optical films adopted in backlights, such as diffuser films and prism films, were used, as shown in Figure 2. The prism film consists of one-dimensional prism grooves collimating the incident light anisotropically. When two prism films were used, denoted as “P2”, one prism film was put over the other, with the directions of the prism grooves being perpendicular to each other.

## 3. Results and Discussion

As a first step, the wall-type LED lighting without any QD caps was investigated for two QD concentrations, 5 and 7.5 wt%, and two reflector types. Appendix A shows the dependence of the correlated color temperature (CCT) of the LED lighting on the reflector property and the QD concentration. As the QD concentration increased from 5 to 7.5 wt%, the CCT decreased due to the larger color conversion efficiency. On the other hand, the CCT decreased when the reflector was changed from the mirror-type to the diffuse-type. In the case of the diffuse reflector, the reflected light was directed in many different directions in the light cavity formed by the bottom and side reflectors and the top PC diffuser. Diffusely reflected light was expected to pass through the QD wall more times than in the case of the mirror reflector. The enhanced color conversion at the higher QD concentration under the condition of the diffuser reflector increased the CRI, as shown in Appendix A. Thus, we will mainly focus on the two conditions, i.e., at the QD concentration of 7.5 wt% and using the diffuser reflector, henceforth. The results obtained from the LED lighting with a mirror reflector will be compared when necessary. However, it should be noted that the optical characteristics of the LED illumination should be investigated as a function of either the thickness or the QD concentration of the QD film, since there should be an optimal condition of color conversion via the remote QD components.

A total of 12 designs were investigated depending on the reflector type, the configuration of the QD film (wall-type and ring-type) and the arrangement of the QD caps. Appendix A shows the categories of the 12 designs together with the corresponding photos. “Hybrid 1~4 (only QD films)” configurations denote the cases where only QD films were adopted, without any red QD caps being used. “Hybrid 5~8 (12 QD caps)” configurations are associated with the cases where 12 QD caps were adopted, as shown in Appendix A. “Hybrid 9~12 (29 QD caps)” configurations are related to the cases where 29 QD caps were arranged, according to Appendix A. 

A total of 12 designs were investigated depending on the reflector type, the configuration of the QD film (wall-type and ring-type) and the arrangement of the QD caps. Appendix A shows the categories of the 12 designs together with the corresponding photos. “Hybrid 1~4 (only QD films)” configurations denote the cases where only QD films were adopted, without any red QD caps being used. “Hybrid 5~8 (12 QD caps)” configurations are associated with the cases where 12 QD caps were adopted, as shown in Appendix A. “Hybrid 9~12 (29 QD caps)” configurations are related to the cases where 29 QD caps were arranged, according to Appendix A. 

Appendix A summarizes the results obtained from the “Hybrid 1~4 (only QD films)” configurations. The luminance values were nearly the same, within 1%. However, the CCT of the ring-type configuration with a mirror reflector showed the highest CCT compared to other configurations (See Appendix A). In this case, the light hitting the QD ring on the mirror reflector tended to be reflected from the tilted mirror reflector surface in the normal direction, without further conversion. On the other hand, the diffuse reflector under the QD ring could redirect the light in different directions and increase the chance of further color conversion, resulting in a lower CCT value. 

Regarding the comparison between the ring-type and the wall-type QD films, the incident light tends to be reflected toward the normal direction in the case of QD ring film, while the QD wall film can be passed by multiply reflected light along the horizontal optical cavity. The thin QD layer of the QD wall was formed on a transparent PET film so that the randomly directed light was more likely to be converted by the QD wall. Thus, the configuration with the QD wall film showed a slightly higher color conversion efficiency, larger CRI values and a smaller CCT. This enhanced color conversion is responsible for the higher CRI values of the wall configuration with a diffuse reflector, as shown in Appendix A.

Figure 3 shows the results obtained from the “Hybrid 5~8 (12 QD caps)” configurations where 12 QD caps were applied together with either a wall- or a ring-type QD film. The overall luminance values became lower when QD caps were used in LED lighting due to the conversion of the green component to the red component, which is less sensitive to the human eye. The luminance was highest for the Hybrid 8 configuration (ring-type QD film + 12 QD caps), where the color conversion was expected to be the highest due to the diffuse reflective property of the reflector and the wall-type configuration that intersected the light trapped in the lighting cavity. This was supported by the lowest CCT, shown in Figure 3b. The improved color conversion via the QD caps increased the Ra above 90 in all configurations, especially in the cases where the diffuse reflector was used (see Figure 3c,d). This may have been due to the diffusely propagating light in the cavity resulting in a more enhanced color conversion efficiency of the QD components. 

These trends were more pronounced in the “Hybrid 9~12 (29 QD caps)” configurations, where 29 QD caps were used together with either a wall- or a ring-type QD film, as shown in Figure 4. These results indicate that using the diffuse reflector is a better choice for increasing the color conversion efficiency of the remote QD component, and that the wall-type configuration is better for higher CRI values than the ring-type configuration. In particular, the Hybrid 11~12 (29 QD caps + a diffuse reflector) configurations had Ra values above 95, which is highly desirable for high color-rendering applications.

The results from the Hybrid 5–12 configurations (12 or 29 QD caps combined with either a wall- or a ring-type QD film) clearly show that the diffuse reflector with QD caps is more favorable for achieving high luminance. Instead of being redirected and escaping along the normal direction via the mirror reflector, the diffuse reflector tends to scatter the light in the cavity and increase the color conversion efficiency via the QD caps/films, resulting in higher luminance. In addition, when the luminance increases, the CCT tends to decrease, and the CRI tends to increase. This is because, when the red component in the spectrum is increased via the red QD films or caps, the color-rendering property is improved, resulting in lower CCT and higher CRI values. Therefore, from the point of view of the color-rendering properties, the Hybrid 11~12 configurations (29 QD caps + diffuse reflector) are the most favorable. Appendix A show all the individual CRI values of the white LED illumination without any QD components, as well as that with 29 QD caps, together with the ring configuration (Hybrid 11 (29 QD caps + diffuse reflector) configuration), respectively. All CRI values increased significantly, especially the R9 associated with the deep-red component, as the Hybrid 11 (29 QD caps + diffuse reflector) configuration was adopted for the white LED lighting. Thanks to the improvement of the R9, the extended CRI Ra could reach ~94, as shown in Appendix A.

In the next step, different combinations of optical films, shown in Table 1, were applied to the top of the PC diffuser to study the effect of the light recycling in the cavity on the optical properties. It is well known that some of the light incident on the microlens arrays is reflected back. For example, about half of the incident light is reflected back from the microprism grooves of the prism film. This type of light recycling plays an important role in improving the luminance of the backlight for LCD applications [43,44]. Figure 5a,b show two examples of the emitting spectra of the white LED illumination of the “Hybrid 3” (only a ring-type QD film) and “Hybrid 11” configurations (29 QD caps and a ring-type QD film), respectively, under six different combinations of optical films on the PC diffuser. The relative intensity of the red peak near 630 nm, which represents the color conversion efficiency of the QD components, changes drastically depending on the combination of optical films.

When there is no QD cap, the peak near 630 nm, shown in Figure 5a, represents the color conversion efficiency of the ring-type QD film. If there is no optical film other than the PC diffuser, the QD film is rarely excited. The QD film is more excited when additional optical films are placed on the PC diffuser because a portion of the light incident on the film is reflected downward and contributes to the excitation of the ring-type QD film. In particular, the QD film is most excited when two prism films are placed on the PC diffuser due to the high light-recycling capability. A large portion of the light travels back and forth between the prism grooves and the bottom reflector, contributing to the excitation of the QD film. Appendix A show the luminance, the CCT, the Ra and the R9, respectively, of the “Hybrid 1~4” (only QD films) configurations, depending on the combination of optical films placed on the PC diffuser. The CCT decreases with a sequence of D → P1 → D-P1 → P2 → D-P2 according to the excitation level of the QD film in the cavity, regardless of the configuration. Along the same sequence, the CRI values increase due to the greater excitation of the red component. However, the difference in the CRI values between the P2 and the D-P2 cases is very small.

Appendix A show the luminance, the CCT, the Ra and the R9, respectively, of the “Hybrid 5~8” (12 QD caps) configurations, depending on the combination of optical films placed on the PC diffuser. The general trend is similar to the case of the “Hybrid 1~4” (only QD films) configurations shown in Appendix A. It should be noted that the CRI values increase significantly as the light recycling effect becomes more pronounced under two prism films. The R9 reached values greater than 94, and thus the Ra became ~96 under two prism films in the Hybrid 7 configuration (diffuse reflector, 12 QD caps, and the ring-type QD film). The CCT could be adjusted in a wide range between 4200 and 3350 K, as shown in Appendix A. It demonstrates that colorimetric properties can be controlled over a wide range by appropriately combining remote QD components and optical films.

Figure 5b shows that the blue and green components of the spectrum are significantly reduced when 29 QD caps are included in the illumination, due to the further excitation of the red QDs in the caps. This trend becomes more pronounced when optical films are placed on the PC diffuser, especially when two prism films are combined and placed on the diffuser. Figure 6a–d show the luminance, the CCT, the Ra and the R9, respectively, of the “Hybrid 9~12” (29 QD caps) configurations, depending on the combination of optical films placed on the PC diffuser. Appendix A shows the same data, but only for the “Hybrid 11” (29 QD caps + diffuse reflector) configuration as one representative example. Luminance is maximized when two prismatic films are placed on the PC diffuser due to their collimating functions. In this case, the viewing angle becomes narrower and the white LED lighting illuminates a smaller solid angle at a higher intensity. Due to the much higher color conversion efficiency of the light recycling process, the CCT was lowered to ~2800 K, as shown in Figure 6b, which is a very warm yellowish light. As shown in Figure 6c, the Ra does not improve significantly for the “Hybrid 9~12” (29 QD caps) configurations, even under the optical films. These configurations already have high enough CRI values above 95. Optical films enhance the red component more, thus disturbing the spectral balance over the visible range, resulting in a lower Ra. However, the enhancement of the red component by the optical films definitely increases the R9 above 97 for some configurations, as can be seen in Figure 6d.

Regarding the data shown in Figure 5 and Figure 6, the angular distribution of the luminance on the prism films should be explained in more detail. It is well-known that the luminance distribution is anisotropic on a prism film due to the one-dimensional nature of the prism grooves, as shown in Figure 2 [44]. Thus, LED lighting is brighter along, for example, the horizontal viewing angle, while it looks darker along the vertical viewing angle. One way to reduce this anisotropic distribution is to use two orthogonal prism films, as was tried for the “P2” configuration. We can use either no prism film, one prism film or two prism films, depending on the application. The LED light without prism films is more suitable for homogeneously illuminating a larger area, while the one with two prism films can be used as a downlight.

Figure 7 show the chromaticity diagram showing the changes in the color coordinates, depending on the optical film combinations, for the Hybrid 3 (no QD cap), 7 (12 QD cap) and 11 (29 QD cap) configurations, respectively. These configurations are characterized by the diffuser reflector and the ring-type QD film. The color coordinates shift to the red region as the number of QD caps increases. In the same configuration, the color coordinates shift to the red region as the number of optical films increases, especially when two prism films are used. The color coordinates of the white LED with the “Hybrid 3” (no QD cap + a ring-type QD film) configuration nearly coincide with the Planckian locus. However, the difference in color coordinates between the ideal Planckian locus and the experimental points increases with the number of QD caps. This is mainly due to the increased red component caused by the QD caps, which pushes the color coordinates to the right lower part of the chromaticity diagram. As shown in Figure 7, the proper application of remote QD components can control the color coordinates in a wide range, thus realizing a different color appearance of white LED illumination.

Finally, we discuss the important aspects of the optimized conditions obtained in the present study from the point of view of optical structures. First, we found that the degree of QD excitation is determined not only by the thickness and QD concentration in the remote QD components, but also by the optical configurations, especially the optical films. This is mainly due to the light recycling process in the optical cavity. The horizontal cavity effect is influenced by the side reflector and the QD films, while the vertical cavity effect is dominated by the bottom reflector and the top optical films. In particular, the light recycling process becomes more substantial when the retroreflective property of the optical film is increased, resulting in higher color conversion efficiency, a lower CCT and higher CRI values. Regarding the reflecting property of the side reflector, the diffusely reflected light tends to pass through the QD components with a higher probability, contributing more effectively to the color conversion. Figure 6 shows that CCT can be controlled over a wide range by simply changing the optical film configuration while maintaining CRI values above 93. Second, the long-term stability of QD components needs to be ensured. Even though we have not measured the long-term stability of the CRI, we would like to mention the aging results, i.e., (1) the percent power of the QD film as a function of time within 1000 h and (2) the relative quantum efficiency of the QD powders at different temperature conditions (room temperature to 85 °C) within 100 days, as reported in [39]. The results showed that both the relative quantum efficiency and the percent power did not degrade within the measurement conditions. It shows the high long-term stability of the fabricated QD powders and remote components, which will ensure the long-term stability of the optical properties of an LED lighting device. However, the effect of blue LED flux on optical properties was not investigated in this study. Protective methods for lengthening the lifespan needs to be studied in more detail [45]. Finally, it should be noted that the spectral characteristics of general lighting and human circadian rhythms, as well as other health effects, need to be studied more carefully. The use of red QDs is an effective way to readjust the relative proportions of blue, yellow and red spectral components. Further investigation in this regard may contribute to the correlation between the light spectrum and human physiology [46,47,48].

## 4. Conclusions

Two remote red QD components, i.e., QD films and QD caps, have been applied to conventional high-power white LED lighting to improve its color-rendering performance. A total of 12 different configurations with six different combinations of optical films were investigated to find the optimum optical structure. Several design rules can be derived from this systematic investigation.

First, the application of red QDs is an effective way to enhance the red spectral component and thus improve the color-rendering performance of conventional white LEDs. The ring-type or wall-type QD film can be used to achieve an Ra over 87. Using the combination of both QD films and QD caps, it was possible to achieve both an Ra and an R9 above 95. 

Second, using the diffuse reflector was more favorable for achieving a high color-rendering performance than using the mirror reflector. The diffuse reflector redirects the reflected light in different directions within the cavity, improving the color conversion efficiency of the red QDs. Thanks to this effect, the CCT was reduced and the CRI values increased with the adoption of the diffuse reflector.

Finally, constructing a strong vertical cavity between the bottom reflector and the top of the LED lighting is an efficient way to improve the color conversion efficiency of red QD components. As the number of optical films increased or two prism films were used, the intensity of the red peak near 630 nm was greatly increased. To adopt optical films to improve the color-rendering performance was more effective for the configurations where fewer QD components were used. If the CRI is already high enough, it is not possible to improve it further by using optical films, because a strong cavity enhances the red component excessively and thus disturbs the spectral balance in the visible range, resulting in a lower Ra.

The present study showed that the proper application of remote QD components combined with an appropriate optical cavity can control the color coordinates of illumination in a wide range, thus realizing a different color appearance of white LED illumination. In addition, a high CRI of over 95 could be achieved due to the sufficient red excitation from fewer QD components and a strong optical cavity effect.

## Figures and Tables

**Figure 1 nanomaterials-13-02560-f001:**
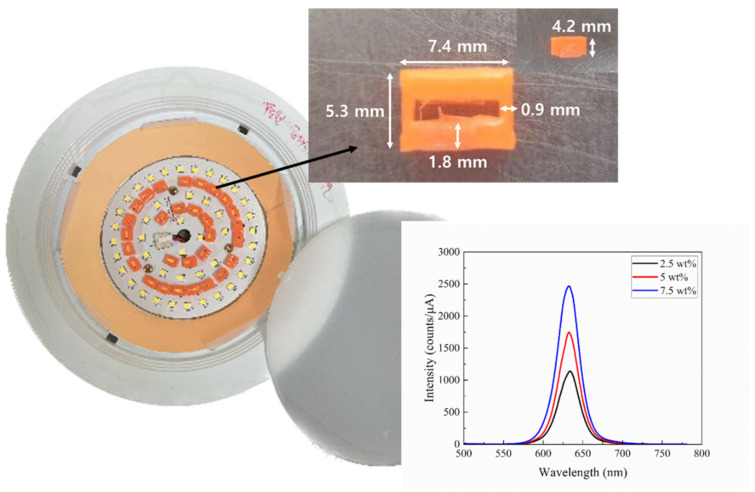
A photograph of the top view of one of the adopted designs, where the conventional white LED fixture can be seen together with remote QD components (the ring-type QD film and the QD caps). The detailed dimensions of the QD cap is also shown. The lower right figure shows the PL spectra of QD films at three concentrations.

**Figure 2 nanomaterials-13-02560-f002:**
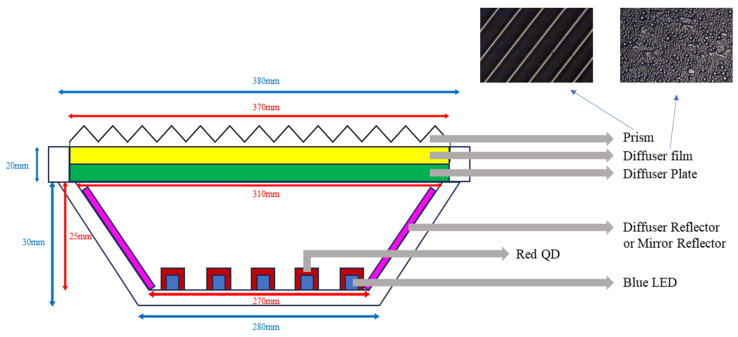
A cross-sectional view of the ring-type white LED lighting along with two microscopic photographs of the diffuser film and the prism film.

**Figure 3 nanomaterials-13-02560-f003:**
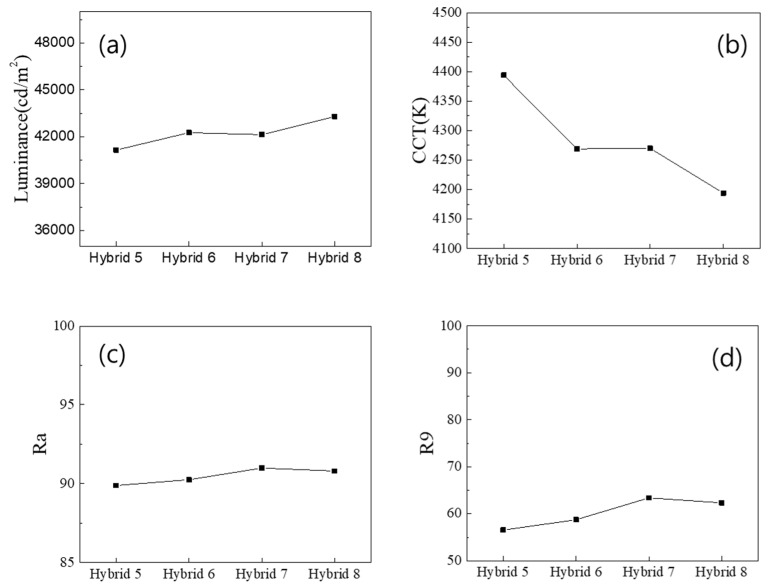
Photometric and color properties of the “Hybrid 5~8 (12 QD caps)” configurations: (**a**) the luminance, (**b**) the CCT, (**c**) the Ra and (**d**) the R9 CRI values.

**Figure 4 nanomaterials-13-02560-f004:**
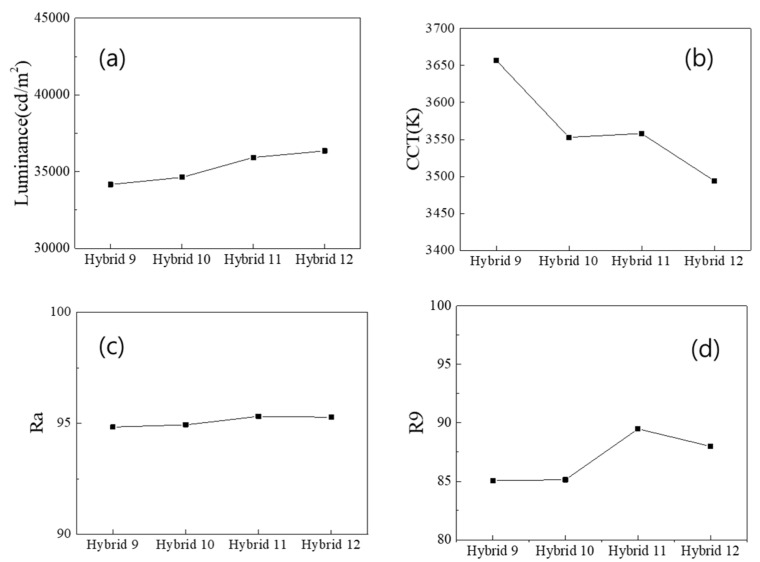
Photometric and color properties of the “Hybrid 9~12 (29 QD caps)” configurations: (**a**) the luminance, (**b**) the CCT, (**c**) the Ra and (**d**) the R9 CRI values.

**Figure 5 nanomaterials-13-02560-f005:**
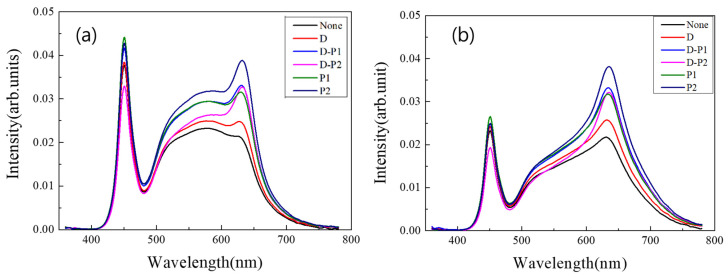
Emitting spectra of (**a**) the Hybrid 3 (only a ring-type QD film) configuration and (**b**) the Hybrid 11 (29 QD caps + diffuse reflector) configuration under six different combinations of optical films.

**Figure 6 nanomaterials-13-02560-f006:**
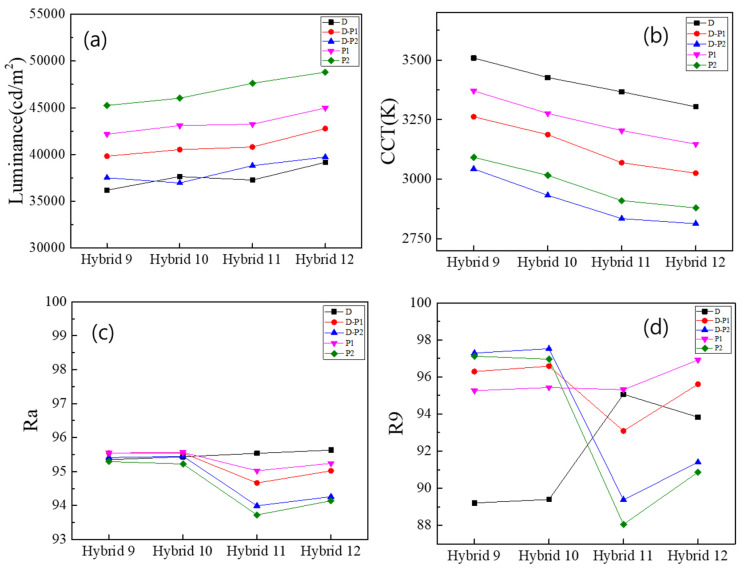
(**a**) The luminance, (**b**) the CCT, (**c**) the Ra and (**d**) the R9 of the “Hybrid 9~12” (29 QD caps) configurations, depending on the combination of optical films placed on the PC diffuser.

**Figure 7 nanomaterials-13-02560-f007:**
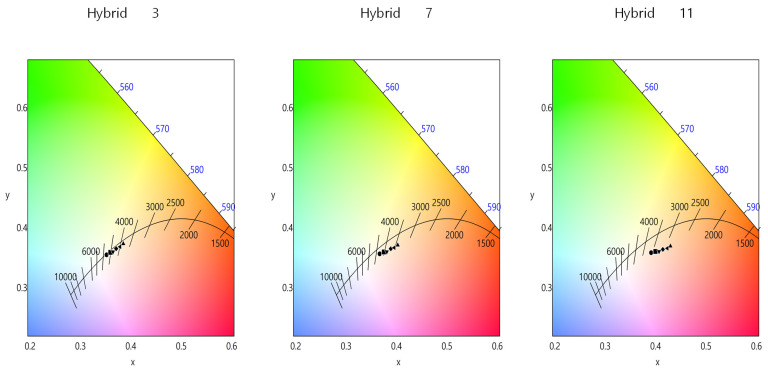
The chromaticity diagram showing the changes in color coordinates, depending on the optical film combinations, for the Hybrid 3 (no QD cap), 7 (12 QD cap) and 11 (29 QD cap) configurations from left to right. These configurations are characterized by the diffuse reflector and the ring-type QD film.

**Table 1 nanomaterials-13-02560-t001:** Dimensions and optical properties of the lighting frame and the PC diffuser plate.

Optical Configuration	Abbreviation	Symbol
None (Only PC diffuser)	None	●
Diffuser film	D	■
Diffuser film + one prism film	D + P1	◆
Diffuser film + two prism films	D + P2	▲
One prism film	P1	▼
Two prism films	P2	◀

## Data Availability

Data presented in this article is available on request from the corresponding author.

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
