# Peer review of "Color Rendering Index over 95 Achieved by Using Light Recycling Process Based on Hybrid Remote-Type Red Quantum-Dot Components Applied to Conventional LED Lighting Devices"

_nanomaterials, 2023, doi:10.3390/nano13182560_

Round 1
Reviewer 1 Report
Please add a scheme explaining a difference between a QD ring film and a QD wall film.
Moreover, it is not clear what are the differnces between different hybrid materials and the differences between them should be clearly indicated in the main text.
What is meant by a two prism film? Is it a one prism film on another?
What are the errors of the measurements of CCT, Luminance, Ra and R9?
Author Response
We highly appreciate the reviewer’s valuable comments.
We revised our manuscript according to the comments as follows. The changes were colored in red in the revised version.
1) Please add a scheme explaining a difference between a QD ring film and a QD wall film. Moreover, it is not clear what are the differences between different hybrid materials and the differences between them should be clearly indicated in the main text.
--> As can be seen from Figures S1(c) and (d) in the Supplementary Materials, the QD ring film was directly laminated on the inclined side reflector while the QD wall film was positioned vertically surrounding the circular white LED array. For clarity, additional photos were included in Table S1(a). In the case of QD ring film, the light incident on it tends to be reflected toward the normal direction, while the QD wall film can be passed by multiply reflected light along the horizontal optical cavity. Thus, the configuration with the QD wall film showed slightly higher color conversion efficiency, larger CRI values and smaller CCT. Thus, we modified the paragraph on the bottom of page 5 as follows:
“Regarding the comparison between the ring-type and the wall-type QD films, the light incident on it tends to be reflected toward the normal direction in the case of QD ring film while the QD wall film can be passed by multiply reflected light along the horizontal optical cavity. The thin QD layer of the QD wall is formed on a transparent PET film so that the randomly directed light is more likely to be converted by the QD wall. Thus, the configuration with the QD wall film showed slightly higher color conversion efficiency, larger CRI values and smaller CCT. This enhanced color conversion is responsible for the higher CRI values of the wall-configuration with a diffuse reflector, as shown in Fig. S3(c)~(d).”
2) What is meant by two prism films? Is it a one prism film on another?
--> A prism film has one-dimensional prism grooves collimating the incident light anisotropically. Thus, when we apply two prism films, the two directions of the two prism films should be orthogonal to each other. This more detailed explanation was included on page 4 just above the Table 1.
“The prism film consists of one-dimensional prism grooves collimating the incident light anisotropically. When two prim films were used denoted as “P2”, one prism film was put over the other the directions of the prism grooves being perpendicular to each other.”
3) What are the errors of the measurements of CCT, Luminance, Ra and R9?
--> They were the order of the symbol sizes shown in Fig. 3, 4, 6, and 7.
Reviewer 2 Report
Baek et al., used the red color emitting QD films and caps on the conventional white light emitting devices illumination and showed an improvement of the color rendering index to over 95%. Further, they discussed the correlation between each of the optical components in achieving higher CRI values of a conventional LED. The authors conducted this work carefully and the results are interesting, but I feel some controls should be added to bring soundness to the work. This work may be publishable after addressing the following technical questions.
1) The distinction between the effects of QD rings and caps is not clearly presented. Are QD rings always present even in the absence of the QD rings? If the answer is yes, it has to be clearly indicated and a blank experiment without a QD ring also should be shown. If the answer is no, one more control experiment is necessary to check the effect of QD rings.
2) What are the effects of QD ring and QD cap thickness on the CRI indexes? It has to be presented.
3) The concentration effects were limited to two concentrations namely 5% and 7.5% despite an increase in Ra values for 7.5% What about higher concentrations? How do the authors decide 7.5% is the right concentration? A higher concentration case should be presented.
4) Prism orientation effects have to be presented to see which orientations are detrimental to the performance of the LED devices.
5) UV-PL spectra of QD should be provided to correlate the results shown in Figure 5.
6) The effect of QD nanoparticle size/shape on the enhancement of CRI has to be investigated.
7) The long-term stability of the estimated CRI indexes should be provided.
Author Response
We highly appreciate the reviewer’s valuable comments.
We revised our manuscript according to the comments as follows. The changes were colored in red in the revised version.
1) The distinction between the effects of QD rings and caps is not clearly presented. Are QD rings always present even in the absence of the QD caps? If the answer is yes, it has to be clearly indicated and a blank experiment without a QD ring also should be shown. If the answer is no, one more control experiment is necessary to check the effect of QD rings.
--> Yes, the hybrid 1~4 configurations include only QD films, without any QD caps, in the ring-type attached on the included side reflector or the wall-type placed vertically surrounding the white LEDs. Therefore, the hybrid 1~4 configurations represent the effect of only the QD film (ring or wall) on the optical properties of the white LED lighting. This point was restated in in the middle part (second paragraph) on page 5 according to this comment as follows.
““Hybrid 1~4(only QD films)” configurations denote the cases where only QD films were adopted without any red QD caps being used.”
The results of the blank experiment without a QD ring was described in “2. Materials and Methods” on page 2 as follows.
“The CRI and the correlated color temperature (CCT) of the original white LED device without any QD components are 82.6 and 5626 K, respectively.”
2) What are the effects of QD ring and QD cap thickness on the CRI indexes? It has to be presented.
--> We could not investigate this parameter on the CRI experimentally due to the limitation of the performance and accuracy of our equipment. By considering this important comment, we added the following sentence at the end of the first paragraph on 5.
“However, it should be noted that the optical characteristics of the LED illumination should be investigated as a function of either the thickness or the QD concentration of the QD film, since there should be an optimal condition of color conversion via the remote QD components.”
3) The concentration effects were limited to two concentrations namely 5% and 7.5% despite an increase in Ra values for 7.5% What about higher concentrations? How do the authors decide 7.5% is the right concentration? A higher concentration case should be presented.
--> It would take much time for growing the QDs, fabricating the QD films and the doing the additional experiments. Instead, we tried to include more discussion about this important comment in combination with our reply to the above comment (2) as follows at the end of the first paragraph on 5:
“However, it should be noted that the optical characteristics of the LED illumination should be investigated as a function of either the thickness or the QD concentration of the QD film, since there should be an optimal condition of color conversion via the remote QD components.”
4) Prism orientation effects have to be presented to see which orientations are detrimental to the performance of the LED devices.
--> It is an important comment. The prism film consists of one-dimensional grooves, thus, it introduces anisotropic luminance distribution. We included the following paragraph on page 9 (the first paragraph):
“Regarding the data shown in Fig.s 5 and 6, the angular distribution of the luminance on prism films should be explained in more detail. It is well known that the luminance distribution is anisotropic on a prism film due to the one-dimensional nature of the prism grooves as shown in Fig. 2 [44]. Thus, the LED lighting is brighter along, for example, the horizontal viewing angle while it looks darker along the vertical viewing angle. One way to reduce this anisotropic distribution is to use two orthogonal prism films as was tried for the “P2” configuration. We can use either no prism film, one prism film or two prism films depending on the application. The LED light without prism film is more suitable for homogeneously illuminating a larger area, while the one with two prism films can be used as a downlight.”
5) UV-PL spectra of QD should be provided to correlate the results shown in Figure 5.
--> The UV-PL spectra of the used QD excited by 365 nm was included in Figure 1 and appropriate explanations were also included in the text and the figure caption on page 3.
6) The effect of QD nanoparticle size/shape on the enhancement of CRI has to be investigated.
--> The size of the QD nanoparticles was intentionally adjusted to ~ 6 nm to meet the requirement of dominant wavelength of ~630 nm. This peak wavelength was chosen because it is preferable from both points of view of efficacy and CRI. If the wavelength is longer than this, the luminous efficiency of the photopic response becomes lower resulting in low efficacy. If the wavelength is shorter than this, the color rendering performance is expected to become worse due to insufficient deep red component. This point was described in the manuscript at the bottom on page 2 as follows:
“The size of the QD nanoparticles was intentionally adjusted to ~ 6 nm to meet the requirement of dominant wavelength of ~630 nm. This peak wavelength was chosen because it is preferable from both points of view of efficacy and CRI. If the wavelength is longer than this, the luminous efficiency of the photopic response becomes lower resulting in low efficacy. If the wavelength is shorter than this, the color rendering performance is expected to be poorer due to insufficient deep red component.”
7) The long-term stability of the estimated CRI indexes should be provided.
--> We have not measured the long-term stability of the CRI, which requires a measurement of a few thousand hours. Instead, we would like to present the aging results, i.e. (1) the percent power of the QD film as a function of time within 1000 hours and (2) the relative quantum efficiency of the QD powders at different temperature conditions (room temperature to 85oC) within 100 days, as reported in Ref. 39. The results showed that both the relative quantum efficiency and the percent power did not degrade within the measurement conditions. It shows the high long-term stability of the fabricated QD powders and remote components, which will ensure the long-term stability of the optical properties of the LED lighting device. This point was included in the manuscript as follows near the bottom of page 10:
“Second, the long-term stability of QD components need to be ensured. Even though we have not measured the long-term stability of the CRI, we would like to mention the aging results, i.e. (1) the percent power of the QD film as a function of time within 1000 hours and (2) the relative quantum efficiency of the QD powders at different temperature conditions (room temperature to 85oC) within 100 days, as reported in Ref. 39. The results showed that both the relative quantum efficiency and the percent power did not degrade within the measurement conditions. It shows the high long-term stability of the fabricated QD powders and remote components, which will ensure the long-term stability of the optical properties of the LED lighting device. However, the effect of blue LED flux on optical properties was not investigated in this study. Protective methods for lengthening the lifetime needs to be studied in more detail [45].”
Reviewer 3 Report
Please see my uploaded review report.

Author Response
We highly appreciate the reviewer’s valuable comments.
We revised our manuscript according to the comments as follows. The changes were colored in red in the revised version.
- The authors provide adequate experimental data to support the concept, but more scientific discussions regarding different configurations should be provided. This will greatly enhance the quality of the paper, especially when delving into the optical studies and insights derived from the findings.
--> Thank you for pointing out this. We prepared one more paragraph on pages 11-12 to include more discussion especially focusing on the optical cavity effect.
- Table S1c requires correction.
--> Thank you for pointing out this mistake. We corrected the numbers.
- The primary content of the paper revolves around color properties under different cavity configurations. To enable a direct comparison, please incorporate the configuration information into the main content, instead of simply using hybrid 1-12.
--> According to this suggestion, we included the configuration information directly after the “hybrid + numbers” over the whole manuscript.
- Since QD is susceptible to the effects of blue light, heat, and humidity, have the authors conducted tests to assess the stability of the QD against water, oxygen, especially high flux blue LED irradiation? These external environmental stimuli could lead to the non-radioactive loss of excitons and severe photo-degradation. To lengthen the lifetime, some protective methods have been discussed in a recent review paper by M. A. Triana, et al. ACS Energy Lett. 7, 1001-1020 (2022). This reference will help readers to know the important lifetime issue of QD devices for lighting and displays.
--> Thank you for this fruitful comment. We did the aging test at high temperatures, the result of which was already reported in our previous publication, Ref. 39. We mentioned this point in the revised manuscript on page 11 as below. However, we did not test our QD against high-flux blue LED irradiation. Instead, we cited the paper mentioned above and included more discussion in the same paragraph:
“Second, the long-term stability of QD components need to be ensured. Even though we have not measured the long-term stability of the CRI, we would like to mention the aging results, i.e. (1) the percent power of the QD film as a function of time within 1000 hours and (2) the relative quantum efficiency of the QD powders at different temperature conditions (room temperature to 85oC) within 100 days, as reported in Ref. 39. The results showed that both the relative quantum efficiency and the percent power did not degrade within the measurement conditions. It shows the high long-term stability of the fabricated QD powders and remote components, which will ensure the long-term stability of the optical properties of the LED lighting device. However, the effect of blue LED flux on optical properties was not investigated in this study. Protective methods for lengthening the lifetime needs to be studied in more detail [45].”
- M. A. Triana, E.L. Hsiang, C. Zhang, Y. Dong, and S.-T. Wu, ACS Energy Lett. 2022, 7, 1001.
- The followings are some relevant papers that readers may find helpful for better understanding this topic and related strategies:
- J.H. Oh, et al. Healthy, natural, efficient and tunable lighting: four-packaged white LEDs for optimizing the circadian effect, color quality and vision performance. Light Sci. Appl. 3, e141 (2014).
- Z. He, et al. Perovskite downconverters for efficient excellent color-rendering, and circadian solid-state lighting, Nanomaterials 9, 176 (2019).
- L. Bellia, et al., Lighting in indoor environments: visual and non-visual effects of light sources with different spectral power distribution, Build. Environ., 46, 1984 (2011).
--> Thank you for the suggestions on these useful papers. They were cited and some more discussion was included on pages 11-12 as follows:
“Finally, it should be noted that the spectral characteristics of general lighting and human circadian rhythms and other health effects need to be studied more carefully. The use of red QDs is an effective way to readjust the relative proportions of blue, yellow, and red spectral components. Further investigation in this regard may contribute to the correlation between the light spectrum and human physiology [46-48].”
- J.-H. Oh, S. J. Yang, and Y. R. Do, Light Sci. Appl. 2014, 3, e141.
- Z. He, C. Zhang, H. Chen, Y. Dong, and S.-T. Wu, Nanomaterials 2019, 9, 176.
- L. Bellia, F. Bisegna, and G. Spada, Build. Environ. 2011, 46, 1984.
Round 2
Reviewer 1 Report
Please correct very minor errors, like lack of space page 5 line 4 and this kind of errors through all the text.
Reviewer 2 Report
The authors clearly replied to the queries and revised their manuscript appropriately for it to be published in Nanomaterials.